# Content and Stability of Hydroxycinnamic Acids during the Production of French Fries Obtained from Potatoes of Varieties with Light-Yellow, Red and Purple Flesh

**DOI:** 10.3390/antiox12020311

**Published:** 2023-01-29

**Authors:** Agnieszka Tajner-Czopek, Elżbieta Rytel, Agnieszka Kita, Anna Sokół-Łętowska, Alicja Z. Kucharska

**Affiliations:** 1Department of Food Storage and Technology, Faculty of Biotechnology and Food Science, Wrocław University of Environmental and Life Sciences, Chełmońskiego St. 37, 51-630 Wrocław, Poland; 2Department of Fruit, Vegetable and Plant Nutraceutical Technology, Faculty of Biotechnology and Food Science, Wrocław University of Environmental and Life Sciences, Chełmońskiego St. 37, 51-630 Wrocław, Poland

**Keywords:** potatoes with different flesh colours, technological stages, French fries, hydroxycinnamic acids, stability, HPLC, antioxidant activity

## Abstract

Potatoes with different flesh colours contain health-promoting compounds, i.e., hydroxycinnamic acids, which vary in content and stability during thermal processing. The aim of this study was to determine the effect of the technological stages of the production of French fries obtained from potatoes with different flesh colours on the content of selected hydroxycinnamic acids, as well as the stability of these acids, their percentage in sum of acids, total phenolic content and antioxidant activity (ABTS, DPPH) in semi-products and ready-to-eat products. During the production of French fries, samples of unpeeled, peeled, cut, blanched, pre-dried and fried potatoes were collected. After peeling, coloured potatoes, especially purple ones, had more hydroxycinnamic (5-CQA, 4-CQA, 3-CQA and CA) acids remaining in the flesh than in the flesh of the light-yellow variety. The greatest losses of the determined hydroxycinnamic acids, regardless of the given potato’s variety, were caused by the stage of pre-drying (about 91%) and frying (about 97%). The French fries obtained from the potatoes with coloured flesh, especially those with purple flesh, had the highest amount of stable 5-CQA and 4-CQA acids as well as 3-CQA acid, already absent in light-yellow French fries. The least stable acid was CA acid, which was not found in any of the ready snacks.

## 1. Introduction

The potatoes (*Solanum tuberosum* L.) with traditional, light flesh have been consumed for a long time due to them having a varied nutrient content (i.e., carbohydrates and proteins). They also contain health-promoting compounds with antioxidant activity (AA), i.e., vitamin C and phenolic acids [1,2,3]. These potatoes can be used for direct consumption after cooking, the production of dried or fried potato snacks, i.e., potato chips and French fries [4,5,6]. Taking into account the health and safety of consumers, as well as the fact that plant raw materials with health-promoting properties (containing valuable biologically active compounds) are increasingly used in food production, the scientists took into account the still not very popular potatoes with colourful flesh (red and purple) [7,8,9,10]. These potatoes show variation in flesh and skin colour [9,11,12,13], as well as in the amount of biologically active compounds [10,14,15]. A variety of polyphenolic compounds, including phenolic acids, are present in the plant raw materials, which have important anticancer, cardio-protective and diabetic nephropathy induced by inflammation prevention effects [15,16,17].

Phenolic acids consist of two groups of compounds: hydroxycinnamic and hydroxybenzoic acids [18,19]. Of these, hydroxycinnamic acids are more abundant and widespread [20,21]. This group includes chlorogenic acid (5-CQA; 5-*O*-caffeoylquinic acid), cryptochlorogenic acid (4-CQA; 4-*O*-caffeoylquinic acid) and neochlorogenic acid (3-CQA; 3-*O*-caffeoylquinic acid), as well as caffeic acid (CA; 3,4-dihydroxycinnamic acid) [21,22,23]. Potatoes of the light and light-yellow fleshed (traditional) varieties, and especially those with coloured flesh, are characterised by a higher content of phenolic acids and contain anthocyanins (absent in traditional potatoes), which can be a good source of valuable compounds with beneficial effects on the human body (e.g., cardioprotective, anti-cancer, anti-inflammatory, anti-diabetic, inhibit oxidative stress and slow down the aging process) [14,15,17,24,25] and should therefore be consumed more often and be destined for food processing.

Depending on the potato variety and the associated flesh colour, different amounts of phenolic acids may be present in the tubers [9,23,24], including derivatives of chlorogenic acid and caffeic acid [26,27,28]. The following acids may also occur in small amounts: p-coumaric acid, ferulic acid, gallic acid, vanillic acid and sinapic acid [14,26,29]. The varying amounts of phenolic acids and their distribution in potato tubers depend, not only on the genotype (variety), but also on the cultivation and climatic factors and storage conditions of the raw material [2,26,29,30,31]. Changes in the amount of phenolic acids in potatoes with different flesh colours may also occur during the technological processing of the raw material [10,12,14,27,32,33]. During the production of potato snacks in the form of French fries, potatoes are subjected to many stages of the technological process, i.e., peeling, cutting, blanching, pre-drying and frying [6,34], which can have a significant impact on changes in the amount of various compounds found in potatoes and in ready products obtained from them [27,32,35,36].

Within the group of potato products, French fries are very popular among a wide range of consumers, especially among young people and children, due to their, among others: attractive taste and smell, varied product range and and quick preparation for eating [34,37]. The increasing consumption of potato snacks may also contribute to an increased interest in potatoes with coloured flesh. Not only because of the unusual colour of the raw material, from which attractive “coloured snacks” can be obtained, but also because of the presence of phenolic compounds with health-promoting effects.

Taking into consideration the beneficial effects of phenolic acids on the human body and the increasing consumption of potatoes in the form of fried snacks, it is important to control the content of these compounds, especially the dominant ones in terms of quantity. It is also important to know what proportion of these acids is discarded with the peel (after peeling the tubers) and what proportion remains in the flesh, as traditionally French fries are created from peeled potatoes [4,6,36]. Another important aspect is the change in the content of phenolic acids and their antioxidant activity, during the production of French fries, as well as the amount of these compounds remaining in the ready product, which may have an impact on improving the “healthiness” of these snacks.

There is no information in the scientific literature on the change in the content of hydroxycinnamic acids, the stability of these compounds and the percentage in the sum of these acids, as well as the amount of total phenolic content (TPC) and antioxidant activity (AA) in samples collected at different stages of the technological production of French fries created using potatoes of varieties with different flesh colours.

In the research presented here, two varieties of the potatoes Mulberry Beauty (with red flesh) and Double Fun (with purple flesh) were used, which are relatively new. However, studies conducted in our Department of Food Storage and Technology for several years show varying contents of the same compounds in potato tubers of a given variety over the years (unpublished studies). Hence, the selection of the suitable potato variety with the highest content of these compounds in the tuber, for the production of French fries, is important because it can affect the preservation of more of these acids in the ready-to-eat snacks. In addition, during the industrial production of French fries, there are losses of the content of various components in potato tubers, including valuable compounds (such as phenolic acids). Since it is possible to modify individual technological stages in the selection of process parameters, therefore, information on the loss of hydroxycinnamic acids content during the course of potato snack production may have a practical dimension. They can provide information to the producers of French fries on which stage of the process losses of these compounds occur and in what amount and at which stage, if possible, should they be modified in terms of minimizing losses, thus leading to the preservation of more acids in ready-to eat snacks.

Accordingly, this manuscript undertook a study in which the aim was to determine the effect of the technological stages of the production of French fries obtained from potatoes with different flesh colours on the content of selected hydroxycinnamic acids, as well as the stability of these acids, their percentage in sum of acids, total phenolic content and antioxidant activity (ABTS, DPPH) in semi-products and ready-to-eat products.

## 2. Materials and Methods

### 2.1. Raw Material

The material collected for the study consisted of three potato (*Solanum tuberosum* L.) varieties: light-yellow-fleshed, traditional—Lady Anna (LA), red-fleshed Mulberry Beauty (MB) and purple-fleshed Violet Queen (VQ). The potato tubers with light-yellow flesh used for the study were collected directly from a potato storage located in the potato products factory. The red- and purple-fleshed potatoes were collected from the experimental plots belonging to the testing station of The Central Institute for Supervising and Testing in Agriculture at Přerov on the Labem (The Czech Republic). The potatoes met the requirements for the basic chemical compounds content in the raw material destined for the processing of French fries, e.g., dry matter, starch and the reducing sugar content. All the investigated potatoes originated from the three growing seasons. The samples of all the potato tubers were harvested after reaching full maturity. Mechanically damaged, greened and sprouted potatoes were rejected after harvest. The samples of each potato variety weighed 15 kg.

### 2.2. Samples Preparation from Potatoes

Potatoes of three varieties were used to prepare the French fries. The ready products were prepared by frying the potato strips in rapeseed oil—according to the methodology in a previous article by Tajner-Czopek et al. [35] with some modifications. Additionally, during the production of French fries, after blanching, the potato strips were dried in a drying oven to obtain the dry mass at about 25%. After frying the potato strips, ready-to-eat product (French fries) was obtained. During the processing of French fries, the following samples were picked for analysis: unpeeled potatoes (raw material)—(UP), peeled potatoes—(PP), potato strips—(PS), potato strips after blanching—(PSaB), potato strips after pre-drying—(PSaP-D) and French fries—(FF). The (UP) and (PP) were cut into 1 cm discs, while the (PS), (PSaB), (PSaB) and (PSaP-D) were cut into smaller pieces. All the samples were freeze-dried with the use of a lyophilizer (apparatus of Edwards Modulyo 4KII freeze dryer, West Sussex, UK). The French fries were degreased using diethyl ether solvent in Büchi Extraction System B-811 apparatus (Büchi Labortechnik AG, Flawil, Switzerland) before analysis. All the obtained dry material was ground in an electric mill (FA-5485 SP-742, TZS First, Austria). The samples after ground were packed in tightly closed plastic containers. From the obtained dried and de-fatted samples, extracts were prepared for the determination of investigated compounds.

### 2.3. Chemicals

The standards of phenolic acids, i.e., 5-*O*-caffeoylquinic acid (5-CQA), 4-*O*-caffeoylquinic acid (4-CQA) and 3-*O*-caffeoylquinic acid (3-CQA), were purchased from TRANS MIT GmbH (Giessen, Germany). The caffeic acid (CA; 3,4-dihydroxycinnamic acid) was purchased from Extrasynthese (Genay, France). The folin–Ciocalteu reagent, sodium hydroxide, 6-hydroxy-2,5,7,8-tetramethylchroman-2-carboxylic acid (Trolox), 2,2-azino-bis-3-ethylbenzothiazoline-6-sulphonic acid diammonium salt (ABTS), 1,1-diphenyl-2-picrylhydrazyl radical (DPPH), formic acid and acetic acid were purchased from Sigma-Aldrich Chemical Co. (Steinheim, Germany). The acetonitrile, methanol, and diethyl ether were bought from POCh (Gliwice, Poland). All the reagents were of analytical grade.

### 2.4. Extraction of Phenolic Acids from Potato Samples

The preparation of extracts from the potato samples was performed according the method described previously in article of Rytel et al. [12] and Piñeros-Niño et al. [23] with some modification. The freeze-dried potato samples and the defatted samples of the French fries (1 ± 0.001 g) were extracted using 50% aqueous methanol acidified with 0.1% acetic acid (50/50/0.1 *v*/*v*/*v*). The mixture was prepared in a graduated tube and then the whole was vortexed for 30 s. The tubes were placed in an ultrasonic bath (UM-2, Unitra-Unima Olsztyn, Poland) for 5 min, then transferred to a centrifuge (MPW-351R, Mpw Med. Instruments, Warsaw, Poland) and centrifuged for 10 min (at 10,000 rpm) at 4 °C and the supernatant was collected. The residue was added to the acidified methanol and re-extracted following the same procedure three times. The supernatants were mixed together. Before application into the column, all the samples were cleaned with using the filters (0.45 µm). Finally, the obtained extracts were used for the analysis of phenolic acids using HPLC and for the determination of the total phenolic content (TPC) and antioxidative activities using the spectrophotometric method.

### 2.5. HPLC Analysis of Phenolic Acids

The concentrations of the investigated acids: 3-CQA-(3-*O*-caffeoylquinic acid); 5-CQA-(5-*O*-caffeoylquinic acid); 4-CQA-(4-*O*-caffeoylquinic acid) and CA (3,4-dihydroxycinnamic acid) were performed using HPLC as described earlier by Tajner-Czopek et al. [38]. To obtain standard solutions, the corresponding amounts of stock solutions were diluted with 50% aqueous methanol (*v*/*v*) acidified with 1% HCl. The calibration curve was obtained on six levels of concentration of standard compound (5-*O*-caffeoylquinic acid), with three injections per level. The chromatogram peak areas were plotted against the known concentrations of the standard solutions. The linear regression equations were calculated by the least-squares method. As the correlation coefficients R^2^ were ≥0.999, the relations were considered linear and acceptable for quantifying the compounds. The linear range was: 20–300 ug/mL, λ_det_: 320 nm, calibration curve: y = 1.083x + 0.819 (y-peak area, x-concentration) and correlation coefficient 0.9999.

The column was operated at 30 °C. The detection wavelength was set at 320 nm. The obtained results were expressed as mg of 5-*O*-caffeoylquinic acid equivalents 100 g^−1^ of dry weight (d.w.) of the analysed samples.

### 2.6. Determination of Total Phenolic Content (TPC) and Antioxidants Activity (ABTS and DPPH) in Extracts

The total phenolic content was determined using a spectrophotometric method using Folin–Ciocalteu (F-C), previously described by Gao et al. [39] and after modification described by Nemś et al. [40]. The absorbance was measured using a spectrophotometer (Rayleigh UV-260, BRAIC, Beijing, China), the value of which was read at λ = 765 nm. The antioxidant activity of the extracts of the tested samples was determined based on the ABTS method according to Re et al. [41], after taking into account the modification described by Tajner-Czopek et al. [38]. The DPPH radical scavenging activity in the tested samples was determined based on the method described by Yen and Chen [42], after modification reported by Tajner-Czopek et al. [38]. The absorbance of the samples was measured using a Rayleigh UV-2601 spectrophotometer (BRAIC, China) at a wavelength of 734 nm (ABTS) and 517 nm (DPPH). All the determinations were performed in three replicates. The results obtained from the Folin–Ciocalteu were expressed as mg of gallic acid equivalent per 100 g of dry weight [mg GAE-100 g^−1^ d.w.], while the results of the ABTS and DPPH were expressed as µmol Trolox equivalent per 1 g [µmol TE·g^−1^ d.w.].

### 2.7. Statistical Analysis

The data were analysed statistically using Statistica v. 13.1 software StatSoft: Tulsa, OK, USA [43]. The experimental data were processed using a one-way analysis of variance (ANOVA), while the homogeneous groups were determined using Duncan’s test at a significance level of *p* ≤ 0.05 and the standard deviations (±SD) were estimated. The correlation analysis was performed to determine the strength and nature of the link between the variables. All the determinations were carried out in three technological replicates and the results represent mean values.

## 3. Results and Discussion

### 3.1. Identification of Hydroxycinnamic Acids in Potatoes and Ready-to-Eat Products

The chromatograms of the hydroxycinnamic acids in the investigated samples, measured using HPLC at 320 nm, are shown in Appendix A.

The investigated compounds were identified by their HPLC retention times, elution order, spectra of the individual peaks (UV/Vis), (Appendix A), spectral data and by comparison with the scientific literature ([23,44]). According to the retention time (5.51 min for 3-*O*-caffeoylquinic acid, 8.20 min for 5-*O*-caffeoylquinic acid, 8.55 min for 4-*O*-caffeoylquinic acid and 8.85 min for 3,4-dihydroxycinnamic acid) and elution order, the peaks in these chromatograms (Appendix A) were identified as: neochlorogenic acid (3-CQA); chlorogenic acid (5-CQA), cryptochlorogenic acid (4-CQA) and caffeic acid (CA), which is in agreement with the results of Ezekiel et al. [14], Piñeros-Niño et al. [23] and Friedman et al. [44].

### 3.2. Hydroxycinnamic Acids Content in Potatoes with Different Flesh Colours

The study found that potatoes of the purple-fleshed variety (VQ) contained the highest sum of hydroxycinnamic acids (380.49 mg·100 g^−1^ d.w.), which were about 13% more compared to the raw material with red flesh (MB) and about 69% compared to that with light flesh LA—(120.05 mg·100 g^−1^ d.w.); thus, the potatoes with coloured flesh contained, on average, about 3.0 times more of the acids tested than the raw material with light-yellow flesh (Table 1). Ezekiel et al. [14] report that the content of the total hydroxycinnamic acids in the potatoes with yellow flesh can range from 28.3 to 273.4 mg·100 g^−1^ d.w. and in those with red and purple flesh in the range from 101.1 to 739.6 mg·100 g^−1^ d.w.

While Silveira et al. [27] report that, in potatoes of the light-yellow flesh variety, the amount of these acids can be 89.72 mg·100 g^−1^ d.w., in raw material with red flesh 241.59 mg·100 g^−1^ d.w. and with purple flesh colouring 267.1 mg·100 g^−1^ d.w. Rytel et al. [12], Ezekiel et al. [14] and Silveira et al. [27] report that the phenolic acid content of the potatoes with coloured flesh can be from 1.5 to 4 times higher compared to the potatoes with traditional (light) flesh colour, while Bellumori et al. [28] and Mullinacci et al. [45] report that they can be even more than 10 times higher.

It was also found that, irrespective of the potato variety and its associated flesh colour, 5-CQA was most abundant in the tubers, followed by 4-CQA and 3-CQA, and had the least CA. The potatoes of the variety (VQ) with purple flesh contained the highest amount of each compound, while the potatoes with light-yellow flesh (LA) contained the least. The content of 5-CQA, 4-CQA and CA in the (LA) potatoes was more than 3 times lower and 3-CQA almost 2.5 times lower compared to the raw material with purple flesh (Table 1).

Rytel et al. [12] reported that the amount of neochlorogenic acid was almost 3 times lower in potatoes with light flesh than in those with purple flesh. A similar relationship of results was observed by Bellumori et al. [28].

It was also stated that the amounts of 5-CQA and 4-CQA in the red-fleshed potatoes (MB) were similar to those in potatoes of the cultivar (VQ), while 3-CQA and CA were 2 times less (Table 1). It was also found that the proportion of 5-CQA acid in the total of the acids tested, in the potatoes of the varieties investigated, regardless of the colour of the flesh, averaged more than 80%, which is consistent with the studies of other authors [9,46,47,48].

Navarre et al. [46] and Finotti et al. [47] reported that, in potatoes with different flesh colours, the share of chlorogenic acid in the total phenolic acids is predominant and may represent about 80%, while Payyavula et al. [48] put its contribution at about 90%. Ru et al. [9] reported that the contribution of 5-CQA to total phenolic acids was the highest, ranging from 35.21% to 81.78%, in the potatoes with different flesh colours.

It was stated that, in potatoes, the proportion of the second most abundant acid after 5-CQA, 4-CQA acid, ranged from 8.9% to 8.2% in the raw material (MB) and (LA), respectively, and that of the next most abundant acid, 3-CQA acid, ranged from 7.2% to 3.5% in potatoes of the same cultivars (Figure 1a,b). In contrast, the proportion of CA acid in the total of compounds tested was the lowest, ranging from 2.8% (variety LA) to 4.5% (variety VQ). The potatoes with purple flesh contained about two times as much of this acid, compared to potatoes with red flesh and more than three times compared to the Lady Anna variety (Figure 1a–c).

Navarre et al. [46] and Pineros-Nino et al. [23] report that, of the group of hydroxycinnamic acids found in potatoes, 4-CQA is the next most abundant after 5-CQA. Finotti et al. [47] also report that 4-CQA was more abundant in potatoes than 3-CQA and accounted for about 12% of the total of all acids, while 3-CQA accounted for 5% of the total amount of these acids. Silveira et al. [27] reported that the caffeic acid content of light-fleshed potatoes was about 3.5 times lower compared to the red-fleshed raw material. Albishi et al. [49] report that in purple-fleshed potatoes the amount of caffeic acid was about 0.1 mg 100 g^−1^ d.m, while Ru et al. [9] found no CA in red- and purple-fleshed potatoes, which was explained by varietal variation and different methods of extraction of compounds from potatoes.

### 3.3. Technological Process of French Fries Production

The technological process of the production of French fries can affect changes in the content of various compounds found in the tubers, because, during the process, the potatoes undergo pre-processing (i.e., peeling and slicing) and thermal treatment [35,36,37].

In the presented studies, the analysed raw material, irrespective of the potato variety used and the associated flesh colour, was subjected to the same conditions of the course of production of the French fries and, therefore, it was possible to compare the effects of the different snack production stages on changes in the hydroxycinnamic acid content, its proportion and the stability of compounds, as well as the total phenolic content and antioxidant activity.

#### 3.3.1. The Influence of Peeling Process on Hydroxycinnamic Acids Content in Potato Flesh

The first preliminary stage in the production of French fries was the peeling of the potatoes, which was carried out under the same conditions for each variety. The peel was removed together with the layer of flesh directly underneath, about 1–1.5 mm thick. It was found that the process of peeling potatoes with different flesh colours had an effect on reducing the hydroxycinnamic acid content. In the peeled potatoes of the three varieties, all the acids tested were found (Table 1). The peeled potatoes with red (MB) and purple (VQ) flesh had a higher amount of total hydroxycinnamic acids by about 6.5 times and almost 10 times, respectively, compared to the sample obtained from the (LA) potatoes. It was noted that peeling potatoes with light-yellow flesh contributed to the greatest loss of the sum of the acids tested (by about 78%) and the least after peeling potatoes with purple flesh (by about 34%), compared to the raw material. In contrast, peeling potatoes of variety (MB) reduced the amount of these compounds by less than half, compared to the raw material (Table 1). Therefore, it was concluded that the peels of the light-yellow fleshed potatoes contained the predominant amount of these acids, which were discarded with it during peeling. In contrast, the skins of the potatoes of varieties with coloured flesh (especially purple) contained a lower amount of these compounds, hence more remained in the flesh of peeled tubers (Table 1). Rytel et al. [12] report that the sum of 5-CQA, 4-CQA, 3-CQA and CA acids in peeled potatoes with light-coloured flesh decreased by about 81% compared to the raw material, while it decreased by about 35% on average in the flesh of the purple potatoes.

After peeling the tubers of the potato variety (LA), there was about 95% less CA acid in the flesh, about 78% less 5-CQA acid and about 75% less of its derivatives (4-CQA and 3-CQA) on average, compared to the raw material. It was also found that, in peeled potatoes with coloured flesh (MB) and (VQ), the amount of 5-CQA acid was predominant among the acids tested. Its amount decreased by about 47% and 30%, respectively, compared to the raw material (Table 1). The amount of 4-CQA and 3-CQA acids, in the flesh of red potatoes, decreased by 45% and 55%, respectively, and, in the flesh of the (VQ) potatoes, by 36% and 48%, respectively. Of the acids tested, CA acid was the least abundant in peeled potatoes of the (MB) and (VQ) cultivars, with a decrease of about 88% and about 89%, respectively, compared to the raw material (Table 1). This indicates a significant accumulation of CA acid and a lower content of the other acids in the peeled potatoes of the tested varieties. The process of peeling the potatoes influenced the change in the proportion of hydroxycinnamic acids in the flesh, compared to that in the raw material (Figure 1a–c). It was found that, irrespective of the potato variety, 5-CQA acid remained the highest proportion in the flesh, with its share of the total acids tested, averaging around 86.6% in the coloured potatoes and 81.3% in the flesh of the light-coloured potatoes. The presence of 4-CQA acid was also found, with a comparable proportion in the flesh of peeled light yellow and red potatoes (9.8% and 9.6%, respectively) and a lower proportion in the flesh of purple potatoes (VQ) at 8.3%. In the peeled potatoes of the cultivar (LA), the proportion of 3-CQA in the total acids tested was 7.9% and, in the flesh of the coloured potatoes (VQ) and (MB), 4.4% and 3.1%, respectively. CA acid was also found to contribute the least of the group of acids tested, with 1.0% in the flesh of the light-yellow coloured potatoes (LA) and 0.8% and 0.6% of the total acids in purple and red flesh, respectively (Figure 1a–c).

Potato peeling results in discarded skins [6,34,37], often rich in phenolic compounds [2,9,13,14,24,26,49]. Rytel et al. [12] consider that the method of potato peeling used, including the thickness of the discarded peel, may affect the quantity of phenolic acids remaining in the flesh. Ezekiel et al. [14], Lachman et al. [24], Friedman [26] and Albishi et al. [50] report that significant amounts of phenolic acids, especially caffeic and chlorogenic acids, are found in the potato peel or immediately below the surface, with varying quantities depending on the variety. Lachman et al. [11] report that the amount of chlorogenic acid in potatoes with coloured flesh, after peeling, decreased on average by about 65.7% compared to the raw material.

#### 3.3.2. The Influence of the Cutting Process on the Content of Hydroxycinnamic Acids in Potato Strips

After peeling, the second preliminary stage in the technological production of French fries is cutting, which is completed by rinsing the potato strips in cold water [34,37]. Phenolic compounds, including phenolic acids, are compounds that are soluble in water [12,51,52], so the quantity of acids during the potato cutting and rinsing process can change.

It was stated that cutting the potatoes of varieties with different flesh colours had an effect on lowering the content of the acids tested in the resulting strips. The samples prepared from raw material with coloured flesh contained all the hydroxycinnamic acids tested, while CA acid was no longer found in the one obtained from potatoes with light-yellow coloured flesh (Table 1). The strips obtained from the red-fleshed potatoes were characterised by having about 7 times the content of total hydroxycinnamic acids and those prepared from purple-fleshed potatoes by about 10 times, compared to the strips from the (LA) potatoes. This is explained by the higher content of these acids in the coloured raw material and in the flesh after peeling. It was found that in the posts obtained from the potatoes with light-yellow coloured flesh, the total amount of the tested acids was lost by about 81%, compared to the raw material, and the least by about 31% in the samples obtained from the (VQ) potatoes, while in the samples obtained from the (MB) potatoes, the amount of these compounds decreased by about 54% (Table 1). In the strips prepared from the potatoes with light-yellow flesh, there was the greatest loss in the amounts of the compounds tested, 5-CQA being about 81% less and 4-CQA and 3-CQA acids about 81% and 79%, respectively, compared to the raw material. In the samples prepared from potatoes of the (MB) and (VQ) cultivars, the content of 5-CQA acid decreased by about 53% and 34%, respectively, compared to the raw material, 3-CQA acid by about 58% and 53%, respectively, and 4-CQA acid decreased by about 49% and 40%, respectively. It was observed that CA acid in the posts obtained from the potatoes with coloured flesh was lost by about 98% each compared to the raw material (Table 1). Rytel et al. [12] report that the process of cutting and washing them in water, applied during the production of dried cubes obtained from purple-fleshed potatoes, had the effect of reducing the amount of total phenolic acids by about 43%, with the greatest loss of caffeic acid (97%) compared to the raw material.

The cutting process of the tested potatoes influenced the change in the proportion of hydroxycinnamic acids in the potato strips compared to the raw material (Figure 1a–c). It was found that 5-CQA acid was the most abundant among the analysed compounds in the samples obtained from the potatoes with different flesh colours. Its share of the total acids was the highest, with 86.7% in the strips obtained from the (MB) potatoes, 87.3% in those from the (VQ) potatoes and 82.5% in those prepared from the potatoes with light-yellow flesh (LA). The proportion of 4-CQA was also high in the cut samples, especially in the strips obtained from the red-fleshed potatoes (10%) and in those from the light-yellow fleshed potatoes (9.7%). In contrast, its proportion was lower in strips from the (VQ) potatoes—8.3%. In the strips (LA) from potatoes, the proportion of 3-CQA in the total acids was 7.8% and was higher than in the samples obtained from the potatoes with purple- (4.3%) and red (3.2%)-coloured flesh. The least of the group of acids tested was CA acid. In the strips obtained from coloured potatoes, its proportion was 0.1% of the total acids. In contrast, it was no longer found in the strips obtained from the potatoes with light-yellow flesh (Figure 1a–c).

#### 3.3.3. The Influence of the Blanching Process on the Content of Hydroxycinnamic Acids in Potato Strips

The cutting of the potatoes is followed by the first heat treatment step in the production of French fries—blanching, which consists of immersing the strips in hot water [34,37]. The process of blanching consists of heat treatment applied to peeled and diced or cubed potatoes in hot water of the temperature 60–85 °C or in steam [6,53].

It was found that the process of blanching the strips obtained from the potatoes of the varieties under investigation had the effect of significantly reducing the quantity of acids. In the blanched samples obtained from the potatoes with light-yellow and coloured flesh, only 5-CQA, 4-CQA and 3-CQA acids were already present. The strips prepared from raw material with red (MB) and purple (VQ) flesh were characterised by a higher amount of total hydroxycinnamic acids, almost 10 times and 12 times, respectively, compared to the sample obtained from the (LA) potatoes. It was noted that blanching the strips prepared from the potatoes with light-yellow coloured flesh resulted in about a 95% loss in the sum of the acids tested, while those from potatoes with coloured flesh averaged about 82% compared to the raw material (Table 1). It was found that, in the blanched sample obtained from the (LA) potatoes, the quantity of 5-CQA, 4-CQA and 3-CQA acids decreased by about 90%, 95% and 97%, respectively, compared to the raw material. In the blanched strips obtained from the potatoes with coloured flesh, CA acid, which was the least of the acids in the raw material, was no longer found and most of it was discarded with the skin (Table 1). This acid also has a higher susceptibility to temperature among the hydroxycinnamic acid group [11,26,54]. It was found that 5-CQA was most abundant in the blanched strips obtained from the potatoes of the (MB) and (VQ) cultivars, but the quantity of 5-CQA decreased by approximately 82% and 80% (respectively) compared to the raw material. The 4-CQA and 3-CQA acid contents also decreased in these samples, with the posts obtained from the (MB) potatoes with about 86% and 80% less, respectively, and those from (VQ) potatoes about 83% and 85% less, respectively (Table 1).

Blanching the strips obtained from the potatoes with different flesh colours affected the change in the proportion of hydroxycinnamic acids in the samples, compared to their proportion in the raw material (Figure 1a–c). It was found that 5-CQA was the most stable acid in the blanched strips obtained from the potatoes of the three varieties. Its proportion in the samples from the (MB) potatoes was 88.7% and from the (VQ) potatoes was 87.7%, while, in the samples prepared from potatoes of the (LA) variety, it accounted for 79.6% of the total acids. In the samples obtained from potatoes of the varieties studied, 4-CQA acid was also stable, especially in the blanched strips obtained from potatoes with light-yellow flesh, where its proportion was 16.1%. In the samples prepared from the potatoes with coloured flesh, the proportion of this acid ranged from 7.2% (MB) to 7.8% (VQ). In all the samples analysed, the proportion of 3-CQA acid was the lowest, averaging around 4.3%. 3-CQA acid was the least stable compound of the group of other acids. It degraded more during blanching in the strips obtained from coloured potatoes than from light-yellow potatoes (Figure 1a–c).

The lower stability of phenolic compounds when exposed to high temperature is confirmed by many authors [5,13,14,26,54,55]. Van Boekel et al. [54] report that the cooking process of various vegetables causes softening, damage to the cell wall and release of compounds contained in it. This includes water-soluble polyphenolic compounds, which, as they pass into the cooking water, can be degraded by the action of temperature. Furia et al. [52] report that the solubility of phenolic acids in water varies and depends on the effect of temperature. Blessington et al. [56] report that, during the heating process (blanching, boiling) of potatoes, there is a migration of phenolic acids from the inner tissue of the tuber to water and a loss in the quantity. This author reports that phenolic acids are thermolabile compounds. Yang et al. [49] noted that cooking potatoes together with the skin has the effect of reducing the loss of valuable components, including phenolic compounds contained in the tubers. This is corroborated by Tian et al. [32], who noted that, in cooked unpeeled potatoes with purple flesh, the amount of chlorogenic acid decreased by 20%, i.e., 80% of its initial content in the tubers still remained. In contrast, Im et al. [57] found a loss of 70% chlorogenic acid in the cooked potato flesh pieces.

#### 3.3.4. The Influence of the Pre-Drying Process on the Content of Hydroxycinnamic Acids in Potato Strips

The next stage of thermal processing in the production of French fries is the drying of blanched potato strips. The purpose of this process is to evaporate water from the semi-product, usually in the range from 1 to 3%, which causes it to be possible to shorten the time of frying the potato pieces in oil [53]. One of the most commonly used pre-drying methods is the convection method, in which energy is supplied along with hot air and the water is removed. However, improperly selected process parameters, including too long a time, may reduce the quality of the potato selvedge [6].

The pre-drying of strips obtained from potatoes with different flesh colours had an effect on reducing the content of the hydroxycinnamic acids studied. 5-CQA and two of its derivatives (4-CQA and 3-CQA) were found in the pre-dried sample obtained from the potatoes of the varieties studied. The strips prepared from the red-fleshed potatoes contained about 11 times the sum of acids and those from potatoes of the variety (VQ) about 16 times compared to the sample obtained from light-fleshed potatoes. The pre-dried strips obtained from the (LA) potatoes had the highest loss of total acids tested (by about 98%) compared to the raw material and the lowest loss of those prepared from potatoes with purple flesh (by about 89%). The samples obtained from the red-fleshed potatoes, on the other hand, contained about 91% less of these compounds compared to the raw material (Table 1). It was also found that, in the pre-dried strips obtained from potatoes of the (LA) variety, the amount of 5-CQA and 3-CQA acids decreased, each by about 98% compared to the raw material, while 4-CQA acid decreased by about 94% (Table 1). In the pre-dried sample obtained from the potatoes of the (MB) variety, the amount of 5-CQA and 4-CQA acid decreased by about 91% and 92%, respectively, and 3-CQA acid by about 95% compared to the raw material. On the other hand, in the pre-dried strips prepared from the purple-fleshed potatoes (VQ), the amount of 5-CQA acid decreased by 87% and 4-CQA and 3-CQA acids by about 92% each compared to the unpeeled potatoes (Table 1).

It was also stated that the pre-drying process had an effect on the further change in the proportion of the acids tested in the strips obtained from the potatoes with different flesh colours, compared to the unpeeled potatoes (Figure 1a–c). In the samples obtained from the potatoes of the investigated varieties, 5-CQA had the highest stability among the acids tested. Its proportion in the potatoes with coloured flesh was 90.1% (VQ) and 89.1% (MB) and, in the sample obtained from the potatoes with light-yellow flesh, it was 71.9%. In contrast, in the pre-dried strips obtained from the (LA) potatoes, 4-CQA and 3-CQA acids showed greater stability than in the samples in potatoes with coloured flesh. The proportion of 4-CQA acid in the samples prepared from (MB) potatoes was 8.3% and this acid was more stable compared to the samples obtained from the (VQ) potatoes—6.0%. In contrast, 3-CQA acid was more stable in the pre-dried samples obtained from the (VQ) potatoes compared to those obtained from the red-fleshed potatoes (Figure 1a–c).

Yang et al. [49] report that drying carried out using different methods, under the action of elevated temperature, has an effect on the change of phenolic compound content in potatoes. Rytel et al. [12] report that the prolonged drying process of potato cubes obtained from raw material with light and purple flesh had a significant effect on the loss of phenolic acids. On the other hand, Chamoarro et al. [58] report that spray drying of potatoes with purple flesh resulted in about a 90% loss of polyphenolic compounds. The authors Chamoarro et al. [58] supposed that the degradation of the compounds was due to both the oxidation process and the effect of the high temperature used during drying.

#### 3.3.5. The Influence of the Frying Process on the Content of Hydroxycinnamic Acids in French Fries

The last stage of thermal processing in the production of French fries, after drying the potato strips, is their frying. Traditionally, the obtained French fries are deep-fried in oil, which is heated to a temperature of 175–180 °C [4,34,35,53].

On the basis of the tests carried out, it was found that frying the strips obtained from the potatoes with light-yellow and coloured flesh had an effect on reducing the amounts of the acids tested. Still, the presence of 5-CQA and 4-CQA acid in all the French fries tested and 3-CQA acid in the samples obtained from potatoes with red and purple flesh (Table 1) were found. The French fries obtained from the potatoes with coloured flesh contained a higher amount of the sum of the acids tested, compared to those obtained from the potatoes with light-yellow flesh, by more than 8 times (MB) and about 14 times (VQ), respectively. In the French fries obtained from the potatoes of the (LA) variety, the total acids decreased by about 99% compared to the raw material, with 1% and 2% of the initial amount of 5-CQA and 4-CQA acid remaining. In contrast, the French fries obtained from the purple-fleshed potatoes (VQ) still contained almost 5% of the initial amount of 5-CQA acid at 14.38 mg·100^−1^ d.w. and there was about 1.6 times more of it compared to the samples collected from the red-fleshed potatoes (MB), in which about 3% of the initial amount of it remained (8.59 mg·100^−1^ d.w.). Additionally, there was almost 15 times more of it compared to the sample fried from the light-fleshed potatoes. In addition, the snacks obtained from the potatoes of the (VQ) variety still contained 3% each of the initial amount of 4-CQA and 3-CQA, while those created from the potatoes of the red-fleshed variety (MB) averaged about 2% of the initial amount of these acids (Table 1).

It was found that the frying process of the pre-dried strips obtained from potatoes with different flesh colours contributed to a change in the proportion of the acids tested in the French fries compared to the raw material (Figure 1a–c). In the snacks prepared from the potatoes with light-coloured flesh, 3-CQA acid was no longer found, which was less stable compared to 5-CQA and 4-CQA acids. In these samples, the share of 5-CQA acid was 83.6% and that of 4-CQA acid was 16.4%, which could indicate their greater stability in the snacks. In contrast, 5-CQA acid was the most stable in the ready-to-eat samples prepared from the red- and purple-fleshed potatoes, with a share of 91.1% in French fries obtained from the (MB) potatoes and 89.9% in the French fries from (VQ). The less stable acids were 4-CQA and 3-CQA, which were found in the French fries obtained from potatoes with purple and red flesh. The share of these two acids was 6.06% and 4.04%, respectively, in the French fries from the (VQ) potatoes and 6.4% and 2.54% in the snacks obtained from the (MB) potatoes with red flesh (Figure 1a–c).

The stability of the phenolic compounds generally decreases when exposed to higher temperatures [5,11,24,26,32,59]. It was, however, Ruitz et al. [60] who showed an increase in the total of phenolic acids (hydroxycinnmic) in the fried chips created from potatoes with coloured flesh, compared to their amount in the raw material. In contrast, Ruitz et al. [60] did not observe an increase in the amount of these compounds after cooking the potatoes. On the other hand, Rytel et al. [12] report that, as a result of prolonged high-temperature drying, the cubes obtained from the potatoes with light-coloured flesh were no longer found to contain phenolic acids, while approximately 4% of their initial amount still remained in cubes obtained from raw material with purple-coloured flesh.

### 3.4. Total Phenolic Content and ABTS and DPPH Activity in Potatoes with Different Flesh Colours

Phenolic acids exhibit significant antioxidant activity, which depends on the reactivity of the phenol moiety that contains a hydroxyl substituent and an aromatic ring. Phenolic acids act as antioxidants (due to the reactivity of phenol moiety—hydroxyl substituent and aromatic ring [21]. Pineros-Nino et al. [23] report that hydroxycinnamic acids, which belong to the phenolic acid group, exhibit health-promoting properties related to their antioxidant activity. The antioxidant activity and the content of the total polyphenols in potatoes varies depending on the cultivar and the associated flesh colour [5,9,32,49,61].

Based on the study, it was found that the highest contents of the total phenolic content and AA were characterised by the potatoes with purple flesh and the samples obtained from them, in the different stages of the production of French fries. The content of the total phenolic content ranged from 1.01 mg GAE·g^−1^ d.w. (variety with LA) to 4.02 mg GAE·g^−1^ d.w. (variety with VQ). In contrast, the total phenolic content of the red-fleshed (MB) potatoes was about 3.5 times higher, compared to the light-yellow fleshed raw material (Table 2).

The quantity of these compounds is similar to the values reported by other authors [27,28,40,61]. Silveira et al. [27] report that the TPC content in the coloured flesh potatoes ranges from 2.880 to 3.241 mg GAE·g^−1^ d.w. and this represents a 2- to 2.5-fold increase compared to the light-fleshed raw material. The higher TPC content in potatoes with coloured (purple and red) compared to the light-fleshed potatoes is also confirmed by Bellumori et al. [28], Jeriene et al. [61] and Deußer et al. [62], who report that the total phenolic content in the potatoes of varieties with different flesh colours ranged from 0.4 mg GAE·g^−1^ d.w. to 5.4 mg GAE·g^−1^ d.w. in the flesh. Nemś et al. [40] reported that in the potatoes with red and purple flesh, the TPC content ranged from 2.1 mg GAE·g^−1^ d.w.to 4.7 mg GAE·g^−1^ d.w. Based on the study, it was found that the potatoes of the variety (VQ) that had the highest total phenolic content also had the highest antioxidant activity. Hence, the ABTS value was almost 3.0 times and the DPPH value was more than 3.4 times higher compared to light-yellow-fleshed potatoes (Table 2).

Silveira et al. [27] report that potatoes of the yellow-fleshed variety show lower antioxidant activity (more than 3 times) compared to the potatoes with purple flesh. The potatoes characterised by higher content of phenolic compounds, including phenolic acids, generally show higher antioxidant activity [5,9,28,59]. Ru et al. [9], Piñeros-Niño et al. [23] and Lachman et al. [24] report that the decreasing content of hydroxycinnamic acids in the samples may have an effect on lowering antioxidant activity.

### 3.5. Total Phenolic Content and Antioxidant Activity of ABTS and DPPH during the Production of French Fries

Based on the study, it was stated that during the technological production of French fries obtained from the potatoes with different flesh colours, the content of the total phenolic and antioxidant activity (ABTS and DPPH) in the samples changed, with each successive stage of snack preparation (Table 2). Kita et al. [5], Siveira et al. [27], Bellumori et al. [28], Tian et al. [32], Yang et al. [49], Nemś et al. [55] and Ruiz et al. [60] report that the total phenolic content and antioxidant activity in the potatoes may change during the use of different potato processing methods.

#### 3.5.1. Peeling

As a result of peeling the potatoes, the peel was removed and, with it, some of the phenolic acids, especially in the potatoes of the (LA) variety. Hence, in the flesh of these potatoes, the amount of TPC was about 46% lower and the activity of ABTS and DPPH was on average more than 1.7 times lower compared to the raw material. On the other hand, when the red (MB) and purple (VQ) potatoes were peeled, some of the acids were rejected with the peel, but there was still a higher amount in the flesh, especially in the potatoes of the (VQ) variety. Therefore, this may have had the effect of slightly reducing the amount of TPC by about 4% and the ABTS and DPPH activities by an average of 1.3 times compared to the raw material of this variety. In contrast, when the red-fleshed potatoes were peeled, the amount of TPC decreased by about 18% and the antioxidant activity expressed as ABTS and DPPH by about 1.3 times and 1.5 times, respectively, compared to the unpeeled potatoes (Table 2).

#### 3.5.2. Cutting

It was stated on the basis of the study that the process of cutting the potatoes had an effect on the reduction in hydroxycinnamic acids in the samples tested, but that these losses compared to those after peeling the potatoes were negligible, especially in the posts obtained from the potatoes with coloured flesh. The TPC content in the posts obtained from purple potatoes was reduced by about 12%, while that from the (MB) potatoes was reduced by about 22% compared to the raw material. The ABTS and DPPH activities of the strips obtained from the purple-fleshed potatoes were found to decrease by 1.6 times and more than 1.7 times, respectively, and, in the samples from the red-fleshed potatoes, by about 2.0 times and about 1.9 times, respectively, compared to the raw material. On the other hand, the TPC content of the strips obtained from the potatoes with light-yellow flesh decreased by more than 48% and the ABTS and DPPH activity decreased in this sample by an average of about 2.3 times compared to the unpeeled potatoes (Table 2).

#### 3.5.3. Blanching

The degradation of CA acid and the reduction in the amount of the remaining hydroxycinnamic acids, after blanching the strips from the potatoes with light-yellow flesh, presumably had the effect of reducing both ABTS and DPPH activities in the samples by about 73% TPC and an average of about 5 times compared to the raw material. In the blanched strips obtained from the potatoes with coloured flesh, a small amount of caffeic acid still remained, but the amount of other hydroxycinnamic acids was markedly reduced, which presumably had the effect of reducing the TPC content in the blanched sample obtained from the (MB) and (VQ) potatoes by about 61% and 55%, respectively, compared to unpeeled potatoes (Table 2). The activities expressed as ABTS and DPPH in the blanched samples obtained from the (VQ) potatoes decreased by about 3.1 and 3.5 times, respectively, while those from the (MB) potatoes decreased by almost 3.5 times (ABTS) and about 4.3 times (DPPH) compared to the raw material (Table 2).

Kita et al. [5] report that the blanching process of light and purple flesh potato slices resulted in a reduction in total polyphenols and antioxidant activity in the resulting chips compared to the unblanched samples.

#### 3.5.4. Pre-Drying

The complete degradation of caffeic acid in the pre-dried strips obtained from potatoes with coloured flesh and the lower content of the hydroxycinnamic acids probably had the effect of reducing the amount of TPC in the samples obtained from the (MB) potatoes by about 82% and from the (VQ) potatoes by about 81% compared to the raw material. In these samples obtained from red-fleshed potatoes, the ABTS and DPPH activity decreased by 5 and 6 times, respectively, and, in those from the (VQ) potatoes, by an average of about 4 times compared to the raw material. In contrast, in the pre-dried strips created from the (LA) potatoes, the amount of TPC decreased by about 79% and the activities expressed as ABTS and DPPH by about 6.5 times and about 7 times, respectively, compared to the unpeeled potatoes (Table 2).

Yang et al. [49] report that the drying method of the potato samples had an effect on shaping antioxidant activity. The above author showed that the lowest antioxidant activity was found in the potatoes dried using the hot air convection method, while the highest was in samples dried using the microwave method.

#### 3.5.5. Frying

It was found that the frying process of strips obtained from the potatoes of different flesh colours affected changes in the total phenolic content (TPC) and antioxidant activity (AA). As a result of the frying process, the TPC content and antioxidant activity decreased in the French fries obtained from the potatoes of the studied varieties compared to the raw material. The TPC content in the French fries obtained from the potatoes with coloured flesh, (VQ) and (MB), decreased on average by about 80% and the ABTS activity from 3.5 times to about 5 times in those with purple and red colouring, respectively, while the DPPH in these samples decreased by about 4 and 6 times, respectively, compared to the raw material. In contrast, in the French fries obtained from the potatoes of the (LA) variety, the amount of TPC decreased by about 79% and the ABTS and DPPH activities by an average of about 6.5 times compared to the unpeeled potatoes. On the other hand, it was stated that the TPC content determined using the Folin–Ciocalteu method in the French fries obtained from the potatoes with light-yellow flesh did not change compared to the pre-dried strips, while the activities expressed as ABTS and DPPH increased slightly but were not statistically significant. A similar relationship was found in the samples obtained from the potatoes with red flesh (MB). In contrast, in the French fries prepared from the purple-fleshed potatoes (VQ), the TPC content increased by about 7%, while the ABTS and DPPH increased by about 8% and more than 6%, respectively, compared to the pre-dried sample (Table 2).

The slight increase in the amount of TPC and antioxidant activity, after frying the samples obtained from the potatoes with purple flesh, can presumably be explained by the fact that this raw material had a higher content of reducing sugars in the tubers than the potatoes with red flesh and especially the light-yellow (LA). The amount of reducing sugars in the raw material (VQ) was 0.15 g 100 g^−1^ f.w., which was about 13% and 33% more sugars, respectively, compared to potatoes of the cultivar (MB) and the one with light-yellow flesh (Table 3). The reducing sugars present in the potato tubers, in addition to the amino acids in them, favour the Maillard reaction [63], which occurs during the frying process carried out at a high temperature [64]. During frying, complex polymeric compounds—melanoidins—are formed, which can exhibit antioxidant properties, such as hydroxymethyl furfural [40].

Blessington et al. [56] reports that thermal processes, i.e., frying and microwaving, can contribute to the antioxidant activity of potatoes. Silveira et al. [27] reported that, in the chips and cubes obtained from potatoes with different flesh colours, the TPC content ranged from 0.772 mg GAE·g^−1^ d.w. to 1.704 mg GAE·g^−1^ d.w., representing a lower amount, from 41% to 65%, compared to the raw material. On the other hand, Ruiz et al. [60] noted that, in the ready-to-eat chips prepared from the potatoes with coloured flesh, the content of the hydroxycinnamic acids and total phenols as well as the antioxidant activity increased compared to the raw material. The above authors [27,60] report that the chlorogenic acids formed as a result of processing may originate from polymeric or oligomeric forms of phenolic compounds, which can be degraded and released under higher temperatures, thus increasing the total phenolic content and AA. Silveira et al. [27] also report that the chips obtained from potatoes with purple and red and white flesh retained a higher amount of these compounds than the products prepared from potatoes with light flesh. Kita et al. [5] report that the chips obtained from purple-fleshed potatoes had significantly higher contents of biologically active compounds, total polyphenols and antioxidant activity (ABTS and FRAP), compared to ready samples prepared from light-fleshed potatoes.

### 3.6. Correlation between Individual Phenolic Acid Content and TPC and Antioxidant Activity

The study also determined the correlation between the content of the hydroxycinnamic acids tested, total phenolic content and the antioxidant activity of ABTS and DPPH in potatoes of light-yellow and coloured flesh varieties and in the samples obtained during the production of French fries (Table 4).

It was stated that the hydroxycinnamic acids (5-CQA, 4-CQA, 3-CQA and caffeic acid) found in the potatoes with light-yellow (LA), red (MB) and purple (VQ) flesh and in the French fries created from them were found to be mostly correlated with TPC and ABTS and DPPH activity. The highest correlation was stated between 5-CQA found in the purple-fleshed potatoes (VQ) and TPC content (r = 0.981 **), as well as between this acid and the ABTS and DPPH activity (r = 0.912 ** and r = 0.927 **, respectively). Additionally, a high correlation was found between the 5-CQA acid contained in the red-fleshed potatoes and the amount of TPC and DPPH activity. A relatively high correlation was also found between the 4-CQA acid contained in the variety (VQ) potatoes and the TPC content and activity expressed as DPPH; r = 0.800 **. A low correlation was found between the CA acid contained in the light and purple-fleshed potatoes and the amount of TPC and DPPH activity. In contrast, there was no correlation between the CA acid contained in the red-fleshed potatoes and the TPC found in them, as well as the ABTS and DPPH activity (Table 4).

In the ready-to-eat French fries created from the purple-fleshed potatoes, the highest correlation was found, between 5-CQA and TPC and the two antioxidant activities. Additionally, a high correlation between these components was shown in the samples fried from potatoes with red flesh (MB). In contrast, no correlation was shown between the 3-CQA contained in the French fries created from the potatoes of the three varieties tested and the amount of TPC and the ABTS and DPPH activities (Table 4).

Yang et al. [49] and Jarienė et al. [61] report that there is a significant correlation between the phenolic acid content of light and coloured flesh potatoes and TPC content and antioxidant activity. Ru et al. [9] report that the phenolic acids contained in the potatoes of light, yellow, red and purple flesh varieties showed a positive correlation with antioxidant activity. Yang et al. [49] noted that the total phenolic content was positively correlated with the DPPH activity. On the other hand, Veiticiene et al. [13] showed a significant correlation between the potato variety, frying temperature of chips and total phenolic content.

The information collected in the manuscript may constitute a valuable compendium of knowledge and a basis for further research in the field of searching for an appropriate potato variety/varieties characterised by the highest content of hydroxycinnamic acids, as well as their greatest stability, particularly during the heat treatment of potatoes used in the production of French fries. Increasing the stability of these compounds during heating may require modification of specific technological steps in the production of snacks, which could be an interesting aspect of future research.

## 4. Conclusions

The potatoes of the variety (VQ) with purple flesh had the highest quantity of individual hydroxycinnamic acids (HA) and their sum, among the analysed raw material. Regardless of the potato variety and the associated flesh colour, 5-CQA was the most abundant acid in the tubers, followed by 4-CQA, 3-CQA and CA. The shares of individual acids in the potatoes with coloured flesh, (VQ) and (MB), were higher compared to those with light-yellow flesh. The raw material with purple flesh (VQ) was also characterised by the highest total phenolic content and showed the highest ABTS and DPPH activity, while the variety potatoes (LA) had the lowest. During the technological course of the production of French fries, each process step had an impact on the loss of hydroxycinnamic acid content, changes in their share in total acids, TPC content and ABTS and DPPH activity, with the greatest changes contributed by the stage of pre-drying the potato strips and frying them. The greatest loss of tested acids was found in the samples obtained from the potatoes with light-yellow coloured flesh (LA) and less in those obtained from the potatoes with coloured flesh, especially with purple flesh (VQ). Peeling, cutting, blanching, pre-drying and frying the potatoes were found to reduce the amount of total (HA) in the samples obtained from the potatoes of the (LA) variety from 78% to 99%, in the samples with (MB) from 48% to 97% and in those with (VQ) from 34% to 96%. The most stable acid found in the ready-to-eat French fries was 5-CQA, while snacks obtained from purple-fleshed potatoes (VQ) contained 5%, from red-fleshed potatoes (MB) 3% and from raw material with light-yellow flesh (LA) 1% of the initial amount of this acid. 4-CQA acid was also fairly stable in the fried samples. In contrast, 3-CQA acid was still retained in the snacks obtained from the raw material with coloured flesh, especially with purple flesh (VQ), but in small amounts. The highest correlation between 5-CQA acid and the TPC content and ABTS and DPPH activity was found in the purple-fleshed potatoes (VQ), as well as between this acid and the TPC content and DPPH activity in the red-fleshed potatoes (MB) and the snacks obtained from them. A high correlation was also shown in the (VQ) potatoes between 4-CQA acid and the TPC content and DPPH activity. The potatoes with coloured flesh, especially those with purple flesh (VQ), are a valuable source of hydroxycinnamic acids, including stable 5-CQA and 4-CQA, and a less stable 3-CQA; therefore, this raw material should be recommended for the production of potato snacks.

## Figures and Tables

**Figure 1 antioxidants-12-00311-f001:**
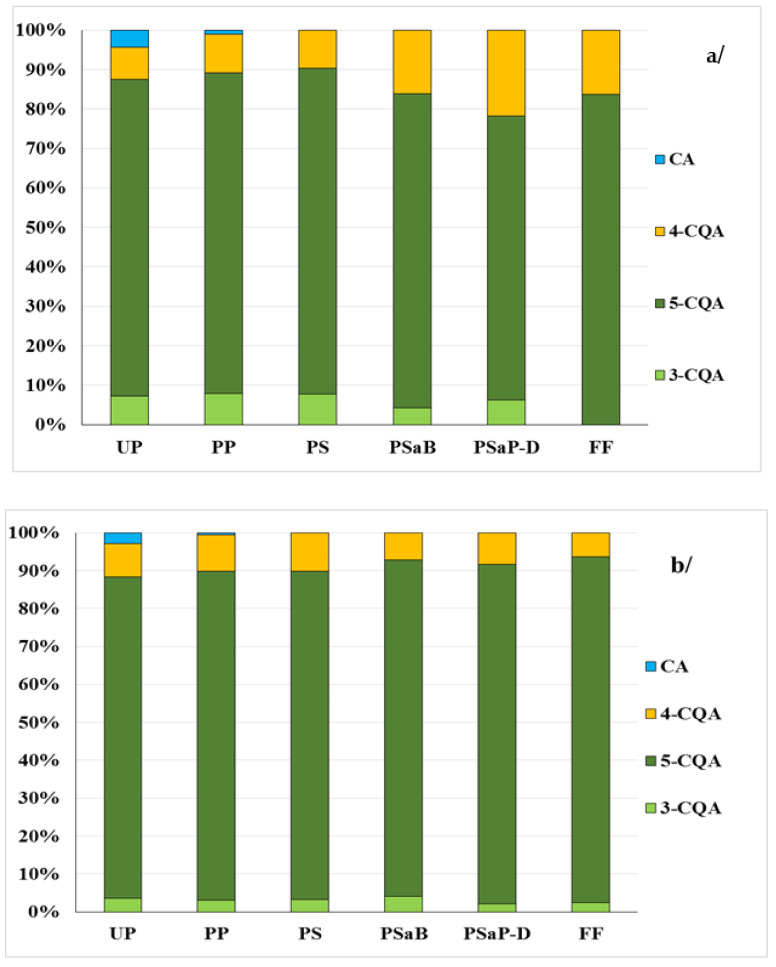
The share of selected hydroxycinnamic acids in their total sum, in the samples obtained during the production of French fries obtained from potatoes: (**a**) (LA)—Lady Anna with light-yellow flesh, (**b**) (MB)—Mulberry Beauty with red-flesh and (**c**) (VQ)—Violet Queen with purple-flesh; 3-CQA (neochlorogenic acid); 5-CQA (chlorogenic acid); 4-CQA (cryptochlorogenic acid); CA—caffeic acid; (UP)—unpeeled potatoes (raw material); (PP)—peeled potatoes; (PS)—potato strips; (PSaB)—potato strips after blanching; (PSaP-D)—potato strips after pre-drying; and (FF)—French fries.

**Table 1 antioxidants-12-00311-t001:** The hydroxycinnamic acids content (mg·100 g^−1^ d.w.) in potato tubers of varieties with different coloured-flesh during the processing of French fries.

Potato	Hydroxycinnamic		French Fries Processing				
Variety	Acids (HA)	(UP)	(PP)	(PS)	(PSaB)	(PSaP-D)	(FF)
(LA)	3-CQA	8.73 ^d^ ± 0.22	2.09 ^c^ ± 0.13	1.74 ^b^ ± 0.10	0.26 ^a^ ± 0.03	0.17 ^a^ ± 0.02	n.d.
Light-yellow	5-CQA	97.11 ^f^ ± 1.28	21.36 ^e^ ± 0.23	18.45 ^d^ ± 0.12	4.85 ^c^ ± 0.12	1.94 ^b^ ± 0.11	0.97 ^a^ ± 0.02
flesh potatoes	4-CQA	9.89 ^d^ ± 0.24	2.57 ^c^ ± 0.11	2.17 ^c^ ± 0.09	0.98 ^b^ ± 0.05	0.59 ^ab^ ± 0.02	0.19 ^a^ ± 0.01
	CA	5.24 ^b^ ± 0.18	0.26 ^a^ ± 0.08	n.d.	n.d.	n.d.	n.d.
Total		120.97 ^e^	26.28 ^d^	22.36 ^d^	6.09 ^c^	2.70 ^b^	1.16 ^a^
(MB)	3-CQA	11.97 ^d^ ± 0.33	5.38 ^c^ ± 0.08	5.02 ^c^ ± 0.08	2.39 ^b^ ± 0.01	0.59 ^a^ ± 0.01	0.23 ^a^ ± 0.02
Red flesh	5-CQA	286.44 ^f^ ± 10.02	149.35 ^e^ ± 9.18	134.62 ^d^ ± 9.25	51.55 ^c^ ± 1.38	25.77 ^b^ ± 0.23	8.59 ^a^ ± 0.12
potatoes	4-CQA	30.01 ^d^ ± 0.28	16.50 ^c^ ± 0.12	15.60 ^c^ ± 1.08	4.20 ^b^ ± 0.02	2.4 ^ab^ ± 0.12	0.60 ^a^ ± 0.05
	CA	9.63 ^c^ ± 0.13	1.06 ^b^ ± 0.04	0.17 ^a^ ± 0.01	n.d.	n.d.	n.d.
Total		338.05 ^e^	172.29 ^d^	155.41 ^d^	58.14 ^c^	28.76 ^b^	9.42 ^a^
(VQ)	3-CQA	21.37 ^e^ ± 0.25	11.11 ^d^ ± 0.81	10.04 ^d^ ± 0.12	3.20 ^c^ ± 0.06	1.70 ^b^ ± 0.16	0.64 ^a^ ± 0.13
Purple	5-CQA	309.55 ^e^ ± 11.22	216.68 ^d^ ± 11.74	204.3 ^d^ ± 10.3	61.91 ^c^ ± 0.67	38.95 ^b^ ± 1.14	14.38 ^a^ ± 0.22
flesh potatoes	4-CQA	32.47 ^e^ ± 1.63	20.78 ^d^ ± 1.13	19.48 ^d^ ± 1.07	5.51 ^c^ ± 0.08	2.59 ^b^ ± 0.08	0.97 ^a^ ± 0.09
	CA	17.10 ^c^ ± 0.98	1.88 ^b^ ± 0.12	0.28 ^a^ ± 0.02	nd.	nd.	nd.
Total		380.49 ^e^	250.45 ^d^	234.1 ^d^	70.79 ^c^	43.24 ^b^	15.99 ^a^

3-CQA (neochlorogenic acid); 5-CQA (chlorogenic acid); 4-CQA (cryptochlorogenic acid); CA—caffeic acid; LA—Lady Anna; MB—Mulberry Beauty; VQ—Violet Queen; (UP)—unpeeled potatoes (raw material); (PP)—peeled potatoes; (PS)—potato strips; (PSaB)—potato strips after blanching; (PSaP-D)—potato strips after pre-drying; (FF) -French fries; d.w.—dry weight; nd.—not detected; Values are represented as mean ± SD—(standard deviation), *n* = 9; ^a–f^—Differences superscripts within rows indicate significant differences (Duncan’s test, *p* ≤ 0.05).

**Table 2 antioxidants-12-00311-t002:** Total phenolic content (mg GAE·g^−1^ d.w.) and antioxidant activity determined using ABTS and DPPH assay (µmol TE·g^−1^ d.w.) in potatoes with different coloured flesh and samples obtained during French fries processing.

	Indicators of Analysis		French Fries Processing				
Potato Variety	Spectrophotometric Measurement	(UP)	(PP)	(PS)	(PSaB)	(PSaP-D)	(FF)
(LA)	TPC	1.01 ^d^ ± 0.06	0.55 ^c^ ± 0.03	0.53 ^c^ ± 0.09	0.27 ^b^ ± 0.05	0.21 ^a^ ± 0.01	0.21 ^a^ ± 0.02
Light-yellow	ABTS	2.90 ^d^ ± 0.05	1.71 ^c^ ± 0.04	1.25 ^c^ ± 0.09	0.58 ^b^ ± 0.02	0.44 ^a^ ± 0.01	0.45 ^a^ ± 0.03
flesh potatoes	DPPH	3.19 ^d^ ± 0.04	1.86 ^c^ ± 0.11	1.34 ^c^ ± 0.02	0.66 ^b^ ± 0.04	0.45 ^a^ ± 0.03	0.46 ^a^ ± 0.01
(MB)	TPC	3.56 ^d^ ± 0.06	2.93 ^c^ ± 0.08	2.78 ^c^ ± 0.04	1.38 ^b^ ± 0.08	0.65 ^a^ ± 0.03	0.68 ^ab^ ± 0.04
Red flesh	ABTS	7.45 ^d^ ± 0.03	5.64 ^c^ ± 0.12	3.77 ^b^ ± 0.05	2.16 ^b^ ± 0.02	1.45 ^a^ ± 0.04	1.50 ^a^ ± 0.01
potatoes	DPPH	9.23 ^e^ ± 0.34	6.23 ^d^ ± 0.06	4.91 ^c^ ± 0.09	2.15 ^b^ ± 0.02	1.53 ^a^ ± 0.10	1.60 ^a^ ± 0.12
(VQ)	TPC	4.02 ^f^ ± 0.10	3.87 ^e^ ± 0.22	3.54 ^d^ ± 0.04	1.81 ^c^ ± 0.09	0.78 ^a^ ± 0.04	0.84 ^b^ ± 0.02
Purple	ABTS	8.12 ^f^ ± 1.03	6.11 ^e^ ± 0.30	5.04 ^d^ ± 0.02	2.58 ^c^ ± 0.02	2.14 ^a^ ± 0.09	2.32 ^b^ ± 0.11
flesh potatoes	DPPH	10.71 ^f^ ± 1.14	7.95 ^e^ ± 0.24	6.12 ^d^ ± 0.09	3.07 ^c^ ± 0.10	2.51 ^a^ ± 0.07	2.68 ^b^ ± 0.12

(LA)—Lady Anna; (MB)—Mulberry Beauty; (VQ)—Violet Queen; (UP)—unpeeled potatoes (raw material); (PP)—peeled potatoes; (PS)—potato strips; (PSaB)—potato strips after blanching; (PSaP-D)—potato strips after pre-drying; (FF)—French fries; TPC—total phenolic content; d.w.—dry weight; ^a–f^—Differences superscripts within rows indicate significant differences (Duncan’s test, *p* ≤ 0.05).

**Table 3 antioxidants-12-00311-t003:** Dry matter, starch and reducing sugars content in potato tubers (fresh weight) of varieties with light-yellow and coloured flesh (mean of years).

Potato Variety	Dry Matter (g 100 g^−1^)	Starch (g 100 g^−1^)	Reducing Sugar (g 100 g^−1^)
(LA) Light-yellow flesh potatoes	22.85 ± 0.11	16.97 ± 0.16	0.10 ± 0.09
(MB) Red flesh potatoes	21.61 ± 0.20	16.40 ± 0.11	0.13 ± 0.10
(VQ) Purple flesh potatoes	21.17 ± 0.14	15.67 ± 0.13	0.15 ± 0.12

(LA)—Lady Anna; (MB)—Mulberry Beauty; (VQ)—Violet Queen; Values are represented as mean ± SD (standard deviation), *n* = 9.

**Table 4 antioxidants-12-00311-t004:** Correlation coefficients (r) for hydroxycinnamic acids, total phenolic content (TPC) and antioxidant activity (ABTS and DPPH) in potatoes with different coloured flesh and samples obtained during the processing of French fries.

Hydroxycinnamic	(LA) Light-Yellow Flesh Potatoes			(MB) Red Flesh Potatoes			(VQ) Purple Flesh Potatoes		
Acids (HA)	TPC	ABTS	DPPH	TPC	ABTS	DPPH	TPC	ABTS	DPPH
Unpeeled potatoes									
3-CQA	0.600 *	n.s.	0.601 *	0.614 **	0.559 *	0.622 **	0.613 **	0.576 *	0.615 **
5-CQA	0.701 **	0.673 **	0.688 **	0.894 **	0.792 **	0.844 **	0.981 **	0.912 **	0.927 **
4-CQA	0.623 **	0.615 **	0.676 **	0.715 **	0.692 **	0.729 **	0.790 **	0.747 **	0.800 **
CA	0.510 *	n.s.	0.509 *	n.s.	n.s.	n.s.	0.524 *	n.s.	0.501 *
French fries									
3-CQA	n.s.	n.s.	n.s.	n.s.	n.s.	n.s.	n.s.	n.s.	n.s.
5-CQA	0.693 **	0.662 **	0.595 **	0.909 **	0.902 **	0.855 **	0.925 **	0.930 **	0.911 **
4-CQA	n.s.	n.s.	n.s.	0.544 *	0.523 *	n.s.	0.632 **	0.628 **	0.613 **
CA	n.d.	n.d.	n.d.	n.d.	n.d.	n.d.	n.d.	n.d.	n.d.

3-CQA (neochlorogenic acid); 5-CQA (chlorogenic acid); 4-CQA (cryptochlorogenic acid); CA—caffeic acid; (LA)—Lady Anna; (MB)—Mulberry Beauty; (VQ)—Violet Queen; n.d.—not detected; n.s.—no significant differences; * and ** indicate significance at (*p* < 0.05 and *p* < 0.01).

## Data Availability

Not applicable.

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
