# Peer review of "Content and Stability of Hydroxycinnamic Acids during the Production of French Fries Obtained from Potatoes of Varieties with Light-Yellow, Red and Purple Flesh"

_antioxidants, 2023, doi:10.3390/antiox12020311_

Round 1
Reviewer 1 Report
This study quantified, comparatively, the contents of HAs during the stages of preparing French fries from selected varieties of sweet potatoes. This work was well-written, and the methods used are robust. I commend the Author's effort for clearly discussing their results. However, their concluding statement should be improved to avoid redundant sentences.
Please check the attached pdf for comments to improve the MS.

Author Response
Dear Reviewer!
The authors are very grateful to the Reviewer for his time and valuable comments and guidance on the manuscript, which increased its merit and were very helpful.
Please find attached your review responses (in pdf.)
Yours sincerely
Agnieszka Tajner-Czopek and co-authors

Reviewer 2 Report
The paper can be published in Antioxidants after the major revision. The authors examined the content of hydroxycinnamic acids and antioxidant activity in raw, semi-products and ready-to-eat products of three potatoes varietes. The topic fits in with the journal's scope. The literature items have all been used in the text. I appreciate the work the authors put into this paper. However, this paper needs a lot of corrections.
My general comment concerns the research done. I would like the authors to clearly justify the purpose of their research. What will be the practical benefits of carrying out all the analyzes at each technological stage, was it not enough to compare the raw material with the final product, and the novelty is to test new varieties? Raw potato cannot be eaten, so the information that it contains more phenolic acids before the next technological stages is of little interest to the average reader. Therefore, I am asking for a thorough justification in the Introduction part of the purposefulness of the research. Detailed notes below:
- Line 33 “Solanum tuberosum” should be italic
- Line 51 “coffee” should be caffeic
- Line 55 “beneficial effects on the human body” - Please describe the effects
- Lines 120-131 and 146 - bold unnecessary
- Line 138 - missing space
- Line 150 “washer” or bath?
- Line 162 please check the name of the compound
- Line 164-165 I do not understand at all why the results are presented as 5-CQA equivalent, if the authors have other standards. This is unacceptable for HPLC analysis. It is known that the surface area is different for each pattern and this way of counting causes distortions. Please make standard curves for each standard separately and count the content of phenolic acids for each acid separately. In the methodology, please describe how the calibration curves were prepared.
- Line 192 “hydroxycinnami”
- Figure 1a-b - There is no part "b" of the figure
- Figure 1 I do not see the need to include chromatograms in the main part of the article. Please consider transferring the chromatograms to Supplementary Materials. The digits signing the peaks are completely shifted and it is not known which peak represents which compound. Please also add spectra of standard substances and compounds determined in extracts in order to verify the correctness of identification. Please also include the retention times in the text
- Table 1 Please explain exactly what the results in table 1 refer to. At first I thought it was raw material, but such results are in table 2, so I don't understand what these results are. In the text, the table is not discussed separately, so it is also impossible to deduce what it refers to.
- The second thing is the caption of the table - table 1 gives the conversion to 100 g f.w., table 2 to 100 g d.m. If the authors wanted to include the content in fresh weight in Table 1, they should clearly indicate this. However, the term d.m. should be corrected on d.w.
- Lines 236-239 Please specify what contents of compounds were obtained by other authors and compare them with your own research
- Lines 260-262 Please discuss why this comparison was made, as evidenced by changes in the proportions of individual phenolic acids and what practical application it has.
- Line 358 “coffeic”
- Line 564 and 568 – “total polyphenols” The authors marked total phenolic content, not just polyphenols. Please unify the terms to match the previously described abbreviation “TP”.
- Line 566 The abbreviation “AA” is used, which is not explained anywhere
- Lines 582-588, 700 Please standardize the units so that all contents of both other authors and your own research are specified in the same units, e.g. mg/100 g d.w.
- Lines 598-599 italic unnecessary
- Line 707 If the abbreviation “AA” stands for antioxidant activity, then the term "content" cannot be applied to it.
- Table 4 Table 4 is not mentioned in the text. What do the results in the table refer to and are the determinations in the fresh mass?
- The authors mention the table in the supplementary materials, but there is no file available.
Author Response

(The authors gave the same response as above.)

Round 2
Reviewer 2 Report
The authors have revised the paper as suggested. The article can be published in its present form. Congratulations to the authors.